# Authentic Romanian Gastronomy—A Landmark of Bucharest's City Center

Ana-Irina Lequeux-Dincă , Mihaela Preda * and Iuliana Vijulie

Faculty of Geography, University of Bucharest, 1. Blv. Nicolae Bălcescu, 010041 Bucharest, Romania;
ana.dinca@geo.unibuc.ro (A.-I.L.-D.); iuliana.vijulie@g.unibuc.ro (I.V.)
* Correspondence: mihaela.preda@geo.unibuc.ro; Tel.: +40-727-784-038

**Abstract:** Gastronomy represents one of the main defining national cultural elements and is essential for shaping territorial identities and for tourism development, attracting both domestic and international tourists. The landscape in the center of Bucharest has gradually changed under the influence of entrepreneurial initiatives within the hospitality industry, showing at present a rather cosmopolitan urban environment. Despite the significant number of international catering units, better adapted to global tastes, Romanian-themed restaurants represent a landmark of the capital city. In this context, our study focuses on the Romanian authentic local gastronomy offered by the themed traditional restaurants in the center of Bucharest as a stimulating factor for different types of consumers. Aiming to answer several research questions, this research has a complex multi-fold methodological approach, appealing to triangulation which gathered, as main analytic methods, mapping, semantic analyses, and text visualisation, and the interview method (originally and appropriately applied for this case study to experienced employees). The main results show a complex gastronomic landscape that gathers various types of restaurants but outlines those with a Romanian ethnic theme in the center of Bucharest. The study of Romanian restaurants' menus reveals elements of authenticity (e.g., traditional dishes and their regional denominations, local rural ingredients, old recipes, and cuisine techniques) as factors of attractiveness for consumers and as competitive advantages in their market. Moreover, interviews with staff representatives outline restaurants' atmosphere, originality, and price–quality ratio of their food as the main attractive elements for both autochtonous customers and tourists and which offer an advantage in the market. The present study may interest multiple stakeholders, focusing on the development and evolution of the hospitality industry in Romania.

**Keywords:** authentic cuisine; Romanian ethnic restaurants; Bucharest city center; semantic analysis

## 1. Introduction

Authentic gastronomy based on traditional dishes and cooking techniques is an essential element of any culture [1] with important and visible implications for several categories of consumers. Autochthonous visitors and international tourists alike are interested in having unique gastronomic experiences during their travels [2].

During the post-communist period, Romania rebranded itself as a tourism destination and displayed a gradual opening towards the international tourism market [3]. The capital city of Bucharest has significantly increased in the late years the number of incoming tourists for business and leisure purposes [4]. Advertising authentic dishes and food products in various ways for different types of consumers [5–7] was part of leisure market segmentation, destination branding, and national image projection strategies. These strategies were similarly developed in different Central and Eastern Europe (CEE) destinations that were preoccupied after 1990 by their socio-economic transition and were in search of economic sustainability elements [8].

Studies and press articles reveal the boost of restaurant units in post-communist Bucharest, which benefited, after 1990, from restaurant and hotel franchises imported from

the western part of Europe. The capital city accounted for an impressive one-third of the restaurant market in Romania [9]. Bucharest, the main economic and social polarizing center in Romania, is one of the most essential business and entrepreneurship hubs and the largest service provider in the country [10], generating a substantial heterogeneous demand for restaurants and consequently developing a greatly diversified, highly concentrated, and competitive restaurant market. The post-COVID period already confirms for Bucharest the rebound of the tourism sector, resilience, and the predicted transformative and adaptive capacity of restaurants after the important shock induced by the sanitary crisis [11,12].

The high demand for authentic Romanian cuisine in Bucharest remained robust during the pandemic. This resilience was thanks to local consumption, the emergence of take-away and delivery options developed by many restaurants during the COVID-19 sanitary crisis, and the promotion of the food quality and safety of traditional farm products advertised by traditional authentic restaurants. These factors, similar to other case studies [13], enforced the robustness and resilience capacity of these Romanian-themed traditional restaurants in the central area of Bucharest.

Despite the importance of this hospitality sector for Romania and Bucharest in particular, there is a lack of accurate statistics and dedicated studies on this topic. Consumers' preference for local authentic gastronomy is of interest to multiple stakeholders, from tourism planners to local authorities or entrepreneurs in the region. It could determine the evolving transformation and continuity of existing traditional restaurants, the possible future appearance of new units of this type, or even complementary offers of traditional dishes in the case of other kinds of restaurant units.

To fill this research gap, our study aims to reveal the Romanian-specific elements of authentic local gastronomy for themed traditional restaurants in the center of Bucharest. Our investigation integrates an exploratory approach and attempts to answer the following research questions:

RQ1—To what extent are Romanian-themed traditional restaurants present in the central area of Bucharest?

RQ2—How is authenticity revealed by the menus of Romanian-themed traditional restaurants in the central area of Bucharest?

RQ3—Which authentic elements offer competitive advantages to their restaurant units, in the opinion of experienced Romanian-theme restaurant employees?

This paper is structured as follows: first, an extended literature review explaining the cultural gastronomic authenticity in the context of rebranded tourism destinations in CEE countries, the emotional consumer behavior of both local and tourist consumers for traditional food, and the elements of authenticity in Romanian ethnic restaurants are presented. Second, the methods and research phases of our exploratory study based on a multi-method approach are explained. Then, the results are presented, divided into several subchapters corresponding to research phases and methods. Finally, the results are discussed and the conclusions made from this study's main results and limitations and further research orientations are presented.

## 2. Literature Review

### 2.1. Traditional Cuisine—A Reflection of Cultural Authenticity and Gastronomic Regional Identity in the Context of Rebranded Post-Communist Destinations

Traditional cuisine is the product of evolving environmental and historical elements reflecting genuine local culture [14], representing a trend countering globalization, and contributing to shaping and restoring geographic identities and resisting homogenization [15,16]. The association of food with places generates symbolic images and meaningful geographies of cuisines and flavors or geographies of food [7,17].

Gastronomic identity represents a promoted behavior among generations addressing the autochthonous population while also affecting newcomers and encouraging a cultural exchange. Today's city dwellers escape their daily routine in a real or imagined manner via so-called "traditional" food. The culinary heritage of rural areas and food consumption

contributes to the construction of identities for urban tourists and inhabitants [18]. The territorial identity of food is maintained through nostalgia and food memory, which play essential roles in reproducing cuisines of any scale. The food, anchored in its territory of origin, memories of its production, and its consumption, is important for maintaining its territorial identity, as domestic consumers "share historical identity" and "a reading of the present through the imaginary of the past" [19] (p. 73).

Scientists demonstrate a strong connection between unique, authentic dishes and their regional character [20,21], as these so-called "authentic" products symbolize the place and culture of a location [22]. Gastronomy, recognized as a tourism resource, led to the definition of European gastronomic heritage, an important element of regional innovation strategy in the European Union defined as the cultural aspect with which tourists most frequently come into contact [23].

Food is an important means by which to sell the local identity culture to tourists [24], to form or enhance the image of destinations, and potentially affect the behavioral intentions of tourists [25,26]. Gastronomy is becoming more frequently utilized by tourism industries as an essential marketing resource and branding instrument [27,28] and plays a pivotal role for some destinations [29] that are capable of generating valuable, complex, memorable tourism experiences [30,31], especially through novelty and authenticity [32]. Authentic gastronomy may be considered as an important reason for travelling, a destination pull factor [33,34], and a powerful marketing tool [35]. Therefore, food consumption at a destination should not be neglected when territories attempt to rebrand and improve tourism development, especially those destinations for which tourism plays an important role in their overall restructuring process [8].

The ex-communist CEE countries have made great efforts in the last decades to increase the volume of incoming international tourists. In an attempt to break with their past, they developed cultural tourism products by promoting and restoring national identities through marketing and rebranding strategies. This gave them a new visuality and the ability to produce appealing tourist images that were distinctive from socialist, repetitive, discursive images and communist propaganda [4,36–38]. If, at the beginning of the 1990s, the ex-post-communist destinations rebranded themselves in an attempt to establish national identities free from any connection with communism in view of EU accession [8], now the geographical (geopolitical?) debate on their "regional" identity seems more and more pertinent as a result of destination maturation and of successful attempts to display tourism products on the neo-liberal market [39].

Post-communist destinations became preoccupied with marketing and branding their distinctive cultural identity [23] as well as shaping and even reinventing a gastronomic identity as a competitive advantage [40,41] by advertising local dishes as brands of authentic products and practices, representative for their national gastronomies [42,43]. In Slovenia's case, gastronomy was advertised as a substantial element of national identity and consequently became a core element of tourism promotional campaigns [44].

Romanian cuisine in relation to tourism demand is a rather neglected topic in the scientific literature, despite the fact that, following regional trends, Romanian-specific gastronomy has been promoted in recent years within tourism packages meant for incoming tourists. Based on slow food procedures, "influenced by different cultures with which it came into contact (Turkish, French, Austrian, Hungarian, Slovak, Russian, etc.)", and "enriched by the diversity of local resources and habits", Romanian gastronomy is, however, still "too little known and promoted abroad" [45] (p. 681).

Traditional gastronomy is hard to define as the primary motivation for incoming tourists visiting Romania and Bucharest because it is related to Romania's diverse agricultural resources and authentic rural products, which are considered to be a solution for mass tourism in SEE countries [46]. Entrepreneurial initiatives at the local level based on existent infrastructure (e.g., ethnic restaurants, markets, wineries, farms) and food events (festivals and exhibitions) valuing traditional cultural resources have recently begun to gradually define autochthonous regional foodscapes in Romania [47].

*2.2. Traditional Food, a Source of Well-Being That Generates Pleasure from Eating and Emotional Behavior for Both Tourists and Local Consumers, Mirroring the Local Culture in Southeast Europe*

Gastronomy is an inextricable part of our daily lives, and eating out is connected to immediate satisfaction and pleasure due to its strong association with well-being, self-construction, and re-definition [48]. More than a necessity and a cultural element, food is connected to well-being, sensory perception, and behavior in food-related experiences, and food is considered a catalyst that adds value to tourists' experiences [29].

Local food is also an essential part of regional cultural heritage that enriches the image of a destination [49], and traditional food has notably been recognized as a medium for cultural expression and interaction [50]. In the last few years, the consumption of local dishes has become a growing phenomenon [51], and many studies aim to surpass the environmental and psychological factors that influence perception and consumers responses to food stimuli [52,53].

Food is explained as a physiological necessity and a social and cultural construct mirroring the local culture, traditions, and natural environment [54]. The emotions evoked by food are essential in predicting consumers' food preferences [55]. This is an important factor for entrepreneurship in themed restaurants, as demonstrated by the Bucharest city center, where these units serve the needs of both international tourists and local residents. The dominance of domestic demand was made clear during the context of the COVID-19 pandemic, with unprecedented lockdowns and dramatic effects on the psychological well-being of the population [56].

Local consumers are attracted by traditional themed restaurants that advertise the origins and authenticity of their products, appealing to consumers' emotions, familiar memories, and nostalgic feelings [57–59]. Studies have shown that autochthonous consumers prefer familiar dishes and are ethnocentric in their support for authentic products [60], as also demonstrated by dietary behavior maintaining large-scale subsistence family farming in Romania [61] and Bucharest foodies in search of multisensory experiences in venues around the Romanian capital city [62]. Both locals and tourists feel the need to connect with local food and to be informed of its provenance, considered a distinctive characteristic of gastronomic products [63]. Traditional gastronomy is based on cultural characteristics and differences determined by various ethno-cultural contexts, sometimes reflected in dishes overlapping symbols and rituals often associated with cultural heritage [64]. Authentic gastronomy represents a culture of food and, therefore, a social, cultural, or spiritual indicator. This is based on unique regional ingredients, reflecting unique environmental characteristics and distinct ethno-cultural aspects [65]. Traditional restaurants display local cuisines, creating a tight connection between the image of a destination's cuisine and tourists' food preferences and satisfaction [66]. Tourists' access to traditional themed restaurants is a significant advantage that allows cultural preservation, while food globalization is a more and more expanded phenomenon [67–69].

*2.3. Defining Authentic Romanian Gastronomy and Romanian-Themed Traditional Restaurants in the Old Center of Bucharest*

Playing the role of a capital city for centuries, Bucharest has long developed its hospitality services and gastronomy, offering real gastronomic experiences and intercultural interactions [27] within its traditional restaurants, which are often located in emblematic historical heritage buildings [70]. These units offer Romanian dishes, advertising authentic bio-products from branded rural areas and following original recipes [71].

As "cultural ambassadors" of a region, ethnic restaurants are connected to ethnic heritage, advertising authenticity as an essential attribute [72,73]. Consumers seek authenticity through traditional and local dishes [22], and traditional cuisine stands out in fulfilling this need. Therefore, local cuisine in the present dynamic tourismscape [74] of Bucharest is a way to discover and experience the lesser known local culture.

From a constructivist perspective, which is clearly related to perception and context [75], ethnic restaurants in Bucharest often appeal to imitation, emphasizing symbolic meanings and replicating original traditional elements found in village areas.

From an objectivist perspective, which reflects authenticity through genuine elements, a restaurant's environment is considered representative of the ethnic origin of the food [76]. The local context and restaurant design may influence the perception of service and help to differentiate their products from various competitors [77]. Although it may be difficult for tourists to evaluate authenticity, as they may not have the experience and technical expertise [77] of lesser known destinations, the objective authenticity relies greatly upon consumer perception and interpretation of visual elements found in themed restaurants in the center of Bucharest [75]. Symbolic reality and staged cultural representations depend very much on the culture, taste, and professional understanding of the owner [76,78]. Consequently, themed Romanian restaurants in the center of Bucharest are closer from the postmodern perspective to the constructivist frame.

Bucharest contains approximately one-eighth of the HoReCa market in Romania, with over 3000 restaurants. These attract a high percentage of foreign tourists, gaining important tourism revenue [9]. Representing a very dynamic market, the number and variety of restaurants have increased in Bucharest in recent years and continue to grow based upon market demand. Themed Romanian restaurants advertising authentic, local cuisine stand out as a landmark of the Bucharest "foodscape" [79], while trying to keep pace with international catering units that offer varied cuisine as well as certain popular traditional Romanian dishes on their menus.

Themed restaurants in the center of Romania's capital city are not accurately counted by the incomplete and hardly accessible official databases [80]. They include old and new units that face taught competition in the market while trying to resist and attract heterogeneous consumers and to adapt and survive global and regional crises (e.g., the COVID-19 pandemic, energy crisis generated by the Russian invasion and the war in Ukraine). Maintaining the originality and authenticity of branded autochthonous dishes could be an advantage but also a great challenge for restaurants in the present context.

All these aspects underline once more the topicality and the need for the present study in its geographic and temporal context.

## 3. Methods

Missing studies, unreliable statistics, and outdated and unclear official classifications of catering units, as well as the logic of the above-expressed research questions, led to extensive field investigations and a multi-method analysis to study authentic gastronomy as a Bucharest landmark.

### 3.1. Place of Research

The present study focused on Bucharest, the capital city of Romania, a tourist destination of Southeast Europe (Figure 1).

Our research focused on the central part of Bucharest, the area called Centrul Vechi (the Old Center), overlapping the old royal courtyard and limited by Unirii Square, Calea Victoriei, I.C. Brătianu Blvd., and Bălcescu Avenue [81,82], as the most frequented area of the city by foreign and domestic tourists, no matter their reason for traveling, but also by residents. A buffer zone of 1.5 km around the historical center was delimited (using Google Earth Pro 7.3.6.9796 and QGIS 3.10.4 software), allowing us to enlarge the sampling needed for the next stages of our analysis. The distance was chosen considering thesurrounding area within walking distance around the Old Center of Bucharest, as other studies defined buffer zones in tourism research as areas where "a tourist attraction can provide benefits and linkages with the surrounding area both as a tourist attraction and supporting tourism activities" [83] (p. 351). This perimeter comprises the highest density of Romanian-themed restaurant units, representing landmarks of food authenticity in the local urban landscape,

and is heavily promoted through recently developed municipality projects (e.g., Urban promenade, Spotlight festival, Zilele Bucureştiului, Spotlight) [84,85].

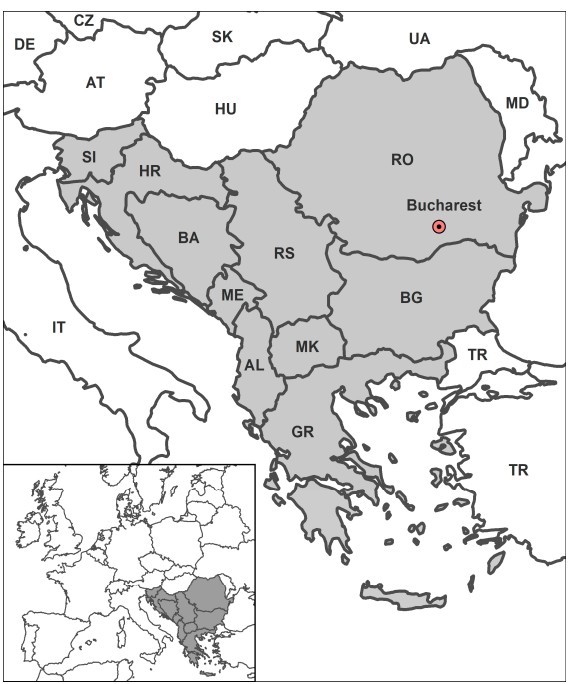

**Figure 1.** Bucharest and its position in Southeast Europe. (Country abbreviations: AT—Austria; AL—Albania; BG—Bulgaria; BA—Bosnia–Herzegovina; ME—Montenegro; GR—Greece; HU—Hungary; HR—Croatia; IT—Italy; MD—Moldova; MK—Macedonia; RO—Romania; RS—Serbia; TR—Turkey; UA—Ukraine).

*3.2. Methods*

From a methodological point of view, a qualitative exploratory and explanatory case study approach was found appropriate to study the topic in its context. This study appealed to triangulation as the various research questions posed by our study demanded a multi-method complementary approach [86] able to ensure the quality of research [79] and avoid biases generated by the effects of singular-source data collection [27]. Triangulation refers to the use of multiple methods and data sources to achieve a comprehensive understanding of phenomena [87,88] and encourages mixing research methods, which may broaden and multiply perspectives on studied topics [89]. This is particularly useful for exploratory research [90] and may have positive implications for hospitality studies involving various stakeholders [28,91].

The threefold methodology is illustrated by the analysis framework below (Figure 2) and refers to:

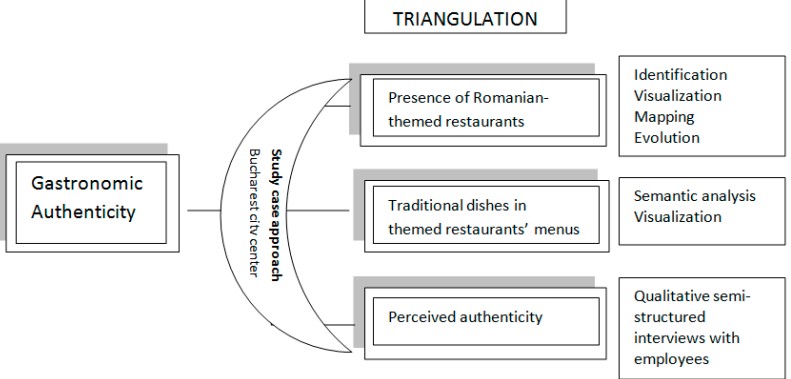

**Figure 2.** The multi-method analysis framework based on complementary data sources.

(a) The illustration and mapping of traditional Romanian restaurants in the Old Center of Bucharest (the above-explained study area), found necessary and appropriate for the exploratory character of our study, was based on field observations to accurately identify unit types and aimed to emphasize the proportion, names, and age of these units as defining elements of the authentic foodscape in the Romanian capital city. Restaurant identification and localization allowed us to underline Romanian-themed restaurants among the other categories of restaurants in the center of Bucharest and further create a heat map (showing areas with concentrations of catering units). A summative synthetic image of restaurant denominations was also created to complement our analysis, and word clouds and statistics were generated with the help of Voyant, an open-source software appropriate for data mining and text analysis [92], also used by other studies on traditional gastronomy [7]. In the end, a table comprising the Romanian-themed restaurants in the studied area and their opening year was created, gathering 22 units that were promoted as restaurants with Romanian-specific cuisine according to the national standardized classification. They served as a basis for the next stages of our research.

(b) The semantic analysis of the 22 menus in the restaurant units that constituted our study sample was the core method of the second research phase in our study. Text was gathered from the online menus available on the internet sites and/or Facebook pages of restaurants in the post-COVID period. Names of main dishes, and consequently of their main ingredients, were analyzed with the help of VOS software 1.6.19 to generate visual maps synthesizing and reflecting semantic clusters of the most frequent words found in the menus. Primarily intended for analyzing bibliometric networks, VOS Viewer is a software tool for creating maps based on network data [93]. VOS Viewer analyzes text files using text mining algorithms to visualize important words and calculate the occurrence of a word or a word stem, often displaying results as clusters of terms [94], also in the case of unstructured text [95]. In our case, an unstructured text was obtained by combining the main words in the menus of the 22 sample units. Data cleaning eliminated linking wording (e.g., "de", "şi", "cu"), declined forms of nouns, and diacritics, particularly used in the Romanian language, to obtain a uniform, plain text containing the main words and avoiding duplicates (e.g., carnat or cârnat). The final text contained words for main courses, starters, and desserts (and eliminated text referring to breakfast, salads, and side dishes) to obtain the best illustration of the foodscape in the center of Bucharest from the menus' perspective. The narrative in the description of dishes, the humorous or particular wording (e.g., diminutives), and the branded origin of products were also elements extracted from the menus, and that completed our analysis, emphasizing the main role of the menus in enhancing the authenticity of restaurants and the fact that attractiveness towards food is also connected to the cognitive styles and cultural patterns of consumers [53].

(c) In-depth qualitative semi-structured interviews [96] with experienced employees of the sample restaurants (working for at least three years in the unit) were the main method employed in the third research phase of our study, which allowed us to reveal the perceived authenticity of Romanian-themed restaurants in Bucharest's old city center. This research stage comprises a triangulation methodological framework that complements previous observational, cartographic, and semantic statistical analyses and represents an original, innovative approach to traditional food studies rather than focusing on consumers' opinions. Previous research studies on gastronomy were preoccupied with customers' perceptions of food image [97], food service [98], and food experience [99,100] as drivers of tourists' satisfaction, neglecting the indirect but more general and objective employees' perspective. The present research addresses this research gap and objectively focuses on the opinions of experienced employees on the authenticity of their restaurant (atmosphere, menus) as a stimulating factor for consumption. This approach seems more appropriate for the explorative character of our study and for the complex, sensitive topic of authenticity, which is hard to capture from the subjective and very heterogenous point of view of various types of consumers (e.g., international and domestic tourists, residents). Face-to-face semi-structured interviews were performed in the post-COVID-19 period (May 2023)

in the 22 sample restaurant units and included questions about the restaurants' themes and names, the ways of incentivizing tourists, the perceived quality of offered services, the traditional dishes, and the success factors for restaurant competitiveness. Informed consent was obtained from all participants attending the interviews and ethical approval for this research objective and its associated interview guide was obtained from the Ethics Committee of the University of Bucharest (document no. 112 /12.12.2023).

Thus, this study provides empirical exploratory results on the authenticity of ethnic Romanian restaurants in the old Bucharest city center by triangulating data provided by in situ observation and the mapping of restaurant units, by semantic analysis of the menus, and by interviews with experienced employees. This could be a starting point for further research studies on traditional gastronomy in Romania, as one of the rare attempts to fill the research gap in the scientific literature for this topic in this region.

## 4. Results

### 4.1. Authentic Gastronomy and Romanian-Themed Restaurants in the Old Bucharest City Center

Overlapping the nucleus from which the development of medieval Bucharest started, the area named the Old Center of Bucharest is located near the ruins of the old royal courtyard and contains numerous heritage buildings, many of which are being reused for entertainment and catering purposes [70,81]. This happened after 1990 and especially after 2000, when restitution laws allowed the transformation of buildings and ownership changes, restoring the traditional hospitality function for this area in Bucharest (Figure 3).

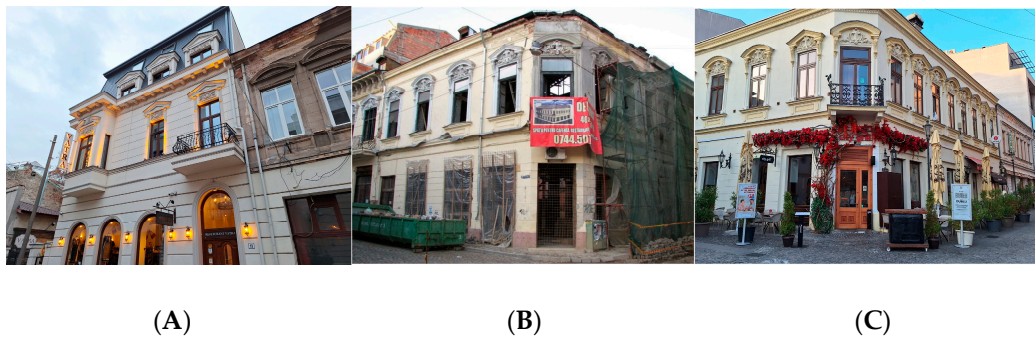

(**A**)            (**B**)            (**C**)

**Figure 3.** (**A**) Vatra restaurant next to a non-renovated building on Brezoianu Street (Source: personal archive); (**B**) The crossroads of Covaci (Blacksmiths) and Șepcari (Hat makers) streets (before renovation). Source: https://turistinbucurestiro (accessed on 6 March 2024); (**C**) The crossroads of Covaci and Șepcari streets (after renovation). (Source: personal archive).

This was abruptly interrupted by the nationalization process during the communist regime, which ignored the rights of the owners and transformed old heritage buildings into residential areas, particularly for economically disadvantaged populations. The area maintained its medieval look, and certain streets still preserve their original names derived from the guilds that were established in the area, crafting and commercializing their products (e.g., Lipscani Street—lipscan: trader who brought his wares from Western Europe; Curtea Sticlarilor/Glassmakers Court, Șelari/Saddlemakers Street, Blănari/Furriers Street, etc.). Few symbolic hospitality heritage places survived and continued their activity uninterrupted (e.g., Hanul lui Manuc/Manuc's Inn, which currently hosts several restaurants and cafeterias; Caru cu Bere, an emblematic restaurant unit in the local foodscape) (Figure 4).

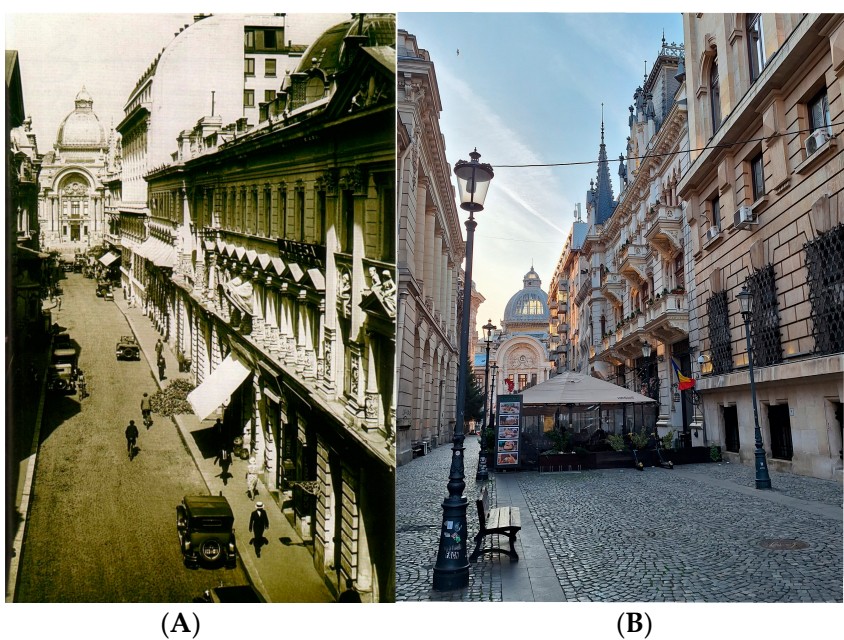

**Figure 4.** (**A**) Stavropoleos Street, view upon Caru' cu Bere restaurant (The Beer Wagon) (1929). Source: https://www.carucubere.ro/istoric/imagini-de-ieri-si-de-azi/ (accessed on 6 March 2024); (**B**) Stavropoleos Street, view upon Caru' cu Bere restaurant (The Beer Wagon) (2024). (Source: personal archive).

The extensive field research in the buffer zone overlapping the Old Center of Bucharest helped us identify and map 227 restaurants of different types (Figure 5A), showing a rather complex, non-homogenous gastronomic landscape in which the global intermingles with the local. As expected, a higher density of restaurants is displayed in the square overlapping the Old Center of Bucharest (Figure 5B). Most of them are international restaurants, and only about 10% of them (22 units) are Romanian-themed restaurants. They further constituted our study sample for the next research phases.

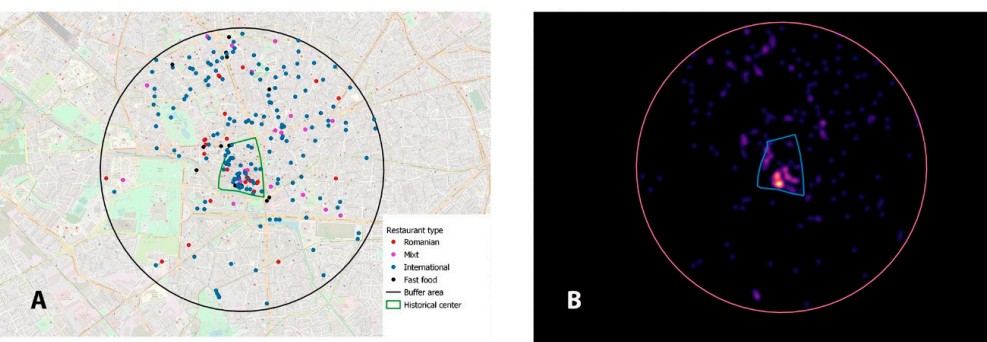

**Figure 5.** (**A**). Restaurant units and their types in the Old Center of Bucharest and in its buffer zone (**left**). (**B**). Heat map showing the concentration of restaurant units in the Old Center of Bucharest and in its buffer zone (**right**). (Source: computed by the authors based on field research data).

Even if they were not the dominant unit type, renowned Romanian-themed restaurants registered as a constant presence in the center of the spatial flow and became a cultural landmark in the very dynamic and continuously transforming "gastroscape" of Bucharest [101]. Often, typical Romanian dishes are proposed by other restaurants in the Bucharest city center rather than the Romanian-themed ones, which try to attract and satisfy various tastes of local and international consumers while fighting the stiff competition of the market. The popularity of Romanian cuisine within this geographical context even

led to the appearance of a particular category of restaurants that combine international and Romanian dishes in an equitable proportion on their menus. Constituting, to a certain extent, promoters of traditional gastronomy, they are called "mixt" in our study and are represented on the map in violet (Figure 5A).

As the restaurant name is one of the first elements that influence customers' authenticity perception [77], stimulating consumption and indicating the type of food and dishes offered by the restaurant, we also analyzed the denominations of the above-mapped units. The obtained graphics and statistics clearly show that the gastroscape in Bucharest is a mix of Romanian and foreign names of catering units. Both autochthonous names (e.g., casa, han, crama, terasa, beraria) and international denominations (e.g., pub, trattoria, taverna, house) indicate a cosmopolitan Bucharest in terms of cuisine with various offers for both local and international consumers. Trattoria, bistro, and pizza (e.g., Latin Pizza, Treevi Pizza) are among the most frequent terms that appear in the name of a restaurant in the center of Bucharest (Figure 6A,B). Certain terms mention popular types of food worldwide (e.g., pizza, BBQ, kebab) but represent at the same time the result of the massive Romanian out-migration flows predominantly oriented towards destinations such as Italy [102] or of the important out-bound flows of Romanian tourists embracing popular dishes from favorite holiday destinations (e.g., Greece and Turkey) [103].

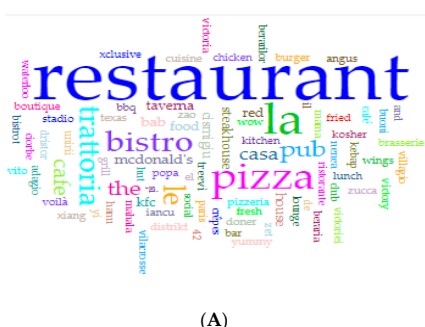

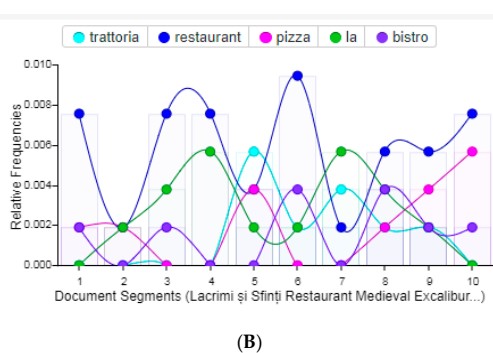

(**A**)  (**B**)

**Figure 6.** (**A**). Word cloud summary based on names of catering units in the Old Center of Bucharest (**left**). (**B**). Relative frequencies and trends of the most frequent terms in the word cloud (**right**).

When analyzing the age of traditional Romanian restaurants in the center of Bucharest, there is an obvious connection between consumers' preferences for autochthonous cuisine and the success of these units. Three restaurants (Hanul lui Manuc (Manuc's Inn), Casa Capşa (Capşa House), and Caru' cu Bere (The Beer Wagon)) date back to the 19th century and became national brands advertising a particular atmosphere and determining consumer loyalty through emotional motivation. Hosted by historical heritage buildings, these units to which Berăria Gambrinus (Gambrinus Brewery) could be added (Table 1) reveal exciting stories from an old Bucharest reminiscent of famous artists and personalities (e.g., the Romanian dramaturgist Caragiale and the famous characters he created).

During the communist period, Romanian restaurants did not diversify as much as they were, without exception, state-owned. The state controlled their numbers and their functioning was often connected to the hotels with which they were associated.

In the early post-communist period, a portion of the previous restaurants were maintained. In contrast, restaurant entrepreneurship, conditioned by the fluctuating economy, multiplied in time, and, encouraged by both international and authochtonous demand, new "authentic" Romanian restaurants designed from scratch with trained personnel and high-quality services appeared next to the old ones (Table 1). During the COVID-19 crisis, consumption in these units was entirely sustained by autochthonous consumers, manifesting important return rates and being emotionally motivated to revisit these units.

**Table 1.** The age of traditional Romanian restaurants.

| Period of Establishment | Restaurant Name | Founding Year |
| --- | --- | --- |
| The 19th century | Hanul lui Manuc | 1812 |
| | Casa Capșa | 1886 |
| | Caru' cu Bere | 1899 |
| The period of the Second World War | Berăria Gambrinus | 1941 |
| The early post-communist period (1990–1999) | Jariştea | 1992 |
| The post-communist period (2000–2023) | La Mama | 2000 |
| | Burebista | 2000 |
| | Vatra | 2001 |
| | City Grill | 2004 |
| | Lacrimi și Sfinți | 2005 |
| | Curtea Berarilor | 2007 |
| | Excalibur | 2008 |
| | Hanul Berarilor | 2009 |
| | La Copac | 2009 |
| | Hanul Hangiţei | 2010 |
| | Casa Gorjană | 2012 |
| | La Nenea Iancu | 2012 |
| | Berăria Nenea Iancu | 2013 |
| | Mahala | 2015 |
| | Taverna Covaci | 2015 |
| | Bodega "La Mahala" | 2015 |
| | Bucătărașul | 2019 |

*4.2. Authentic Traditional Gastronomy in the Old Bucharest City Center*

Based upon the analysis of the elements of authenticity revealed by the menus of the 22 traditional Romanian-themed restaurants located in the central area of Bucharest, the present study outlines certain popular, classical autochthonous dishes as main food attractions, with variants offered on the menus of the catering units. Many restaurants included side dishes in the description of the course as an important added value, underlining autochthonous accompaniments and garnishes. This explains why, through the full counting method using a value 5 for the threshold, the image in Figure 7 was obtained, mainly underlining four clusters.

The cluster in red includes the core part of dishes that could be considered basic popular courses (e.g., ciorbă—sour soup traditionally used as a starter; mititei—grilled ground meat rolls; sarmale—traditional cabbage rolls with ground pork and rice; cârnaţi pleşcoi—sausages branded in Pleşcoi village; porc for pork, present in numerous traditional dishes (Table 2); papanaşi—dessert representing Romanian fried cheese doughnuts), familiar cooking methods (casa—house recipe), or serving methods (e.g., platou—traditional combinations of starters presented on wooden plates and usually including dishes found in the same cluster such as: vinete—eggplant caviar; icre—fish roe salad; zacuscă—traditional vegetable spread; murături asortate—assorted pickles; pastramă de oaie/berbecuţ—smoked sheep pastrama; jumări—Romanian-style pork cracklings) (Figures 8 and 9).

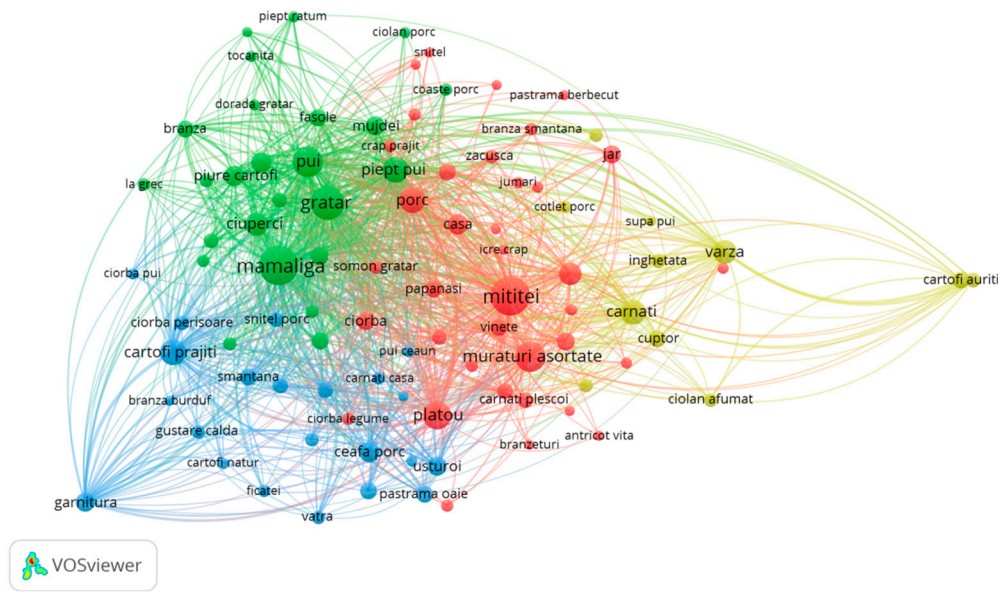

**Figure 7.** Relationship of words in the menus of Romanian-themed restaurants in the center of Bucharest.

**Table 2.** The word count in the menus of Romanian-themed restaurants in the center of Bucharest.

| Term | Occurrence | Term | Occurrence |
|---|---|---|---|
| porc | 104 | cartofi prajiti | 38 |
| gratar | 92 | carnati | 37 |
| ciorba | 85 | varza | 35 |
| mamaliga | 84 | mititei | 18 |
| platou | 61 | mujdei | 16 |

Source: computed by authors.

A secondary cluster (in green) contains the word "mamaliga" as the most important side dish and/or accompaniment for main courses, representing the Romanian polenta and traditionally often replacing bread. According to Scrob [104], mămăligă represented the core food of Romanian villagers for centuries, and a massive switch to the predominant consumption of bread occurred as late as the 1960s. However, dietary family traditions and conservatism still show a hedonic preference for mămăligă, especially for "specific meals based on early dietary experiences" of consumers [104] (p. 234). Other main ingredients for side dishes such as fasole—beans, ciuperci—mushrooms, piure cartofi—smashed potatoes, and their traditional preparation methods (e.g., tocăniță—Romanian stew) are also present in this cluster. Other meats of secondary importance after pork (which is present in the first cluster) such as pui—chicken or the most popular fish dish (e.g., crap prăjit—fried carp) and popular accompaniments (e.g., mujdei—garlic cream) are also included in the second cluster (Figure 10).

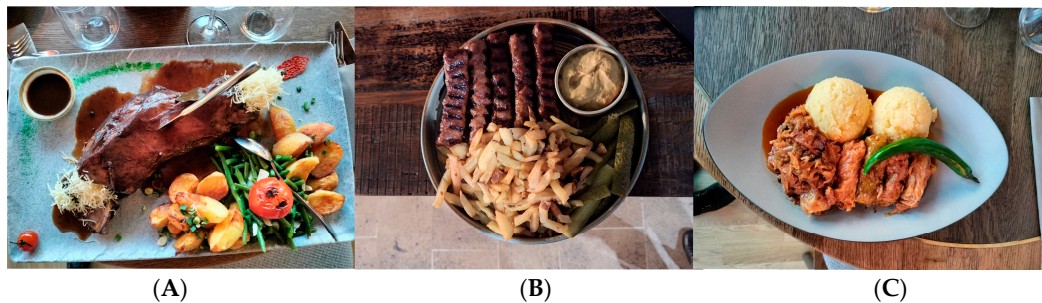

**Figure 8.** (**A**) Caramelized lamb knuckle (Manuc's Inn restaurant) (Source: personal archive); (**B**) Mici with fries and pickled cucumbers (Curtea Berarilor) (Source: personal archive); (**C**) Sarmale (Manuc's Inn restaurant) (Source: personal archive).

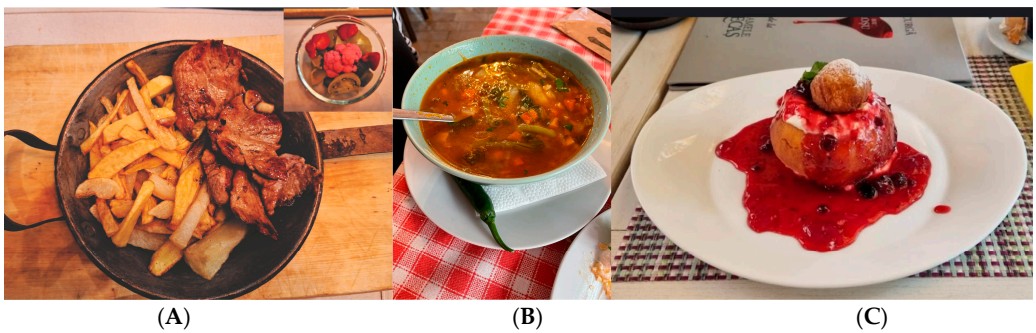

**Figure 9.** (**A**) Pork on a Border with assorted home-made pickles (Lacrimi şi Sfinţi restaurant) (Source: personal archive); (**B**) Ciorbă dish in the Old Center of Bucharest. (Source: personal archive); (**C**) Papanaşi dish in the Old Center of Bucharest. (Source: personal archive).

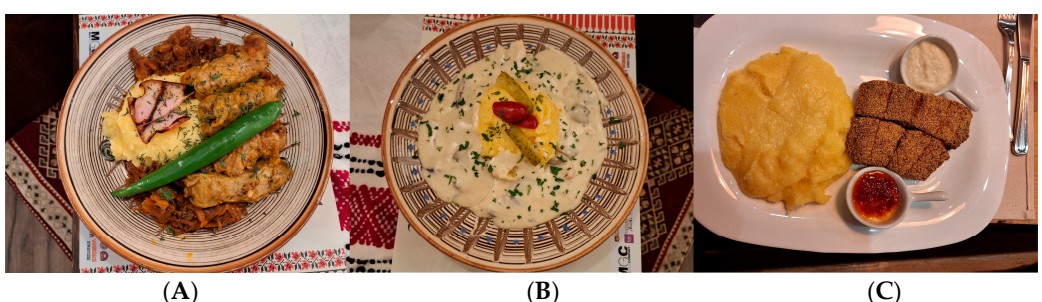

**Figure 10.** (**A**) Sarmale served with Romanian polenta and hot pepper (Taverna Covaci restaurant) (Source: personal archive); (**B**) Creamy mushroom stew with chicken served with Romanian polenta (Taverna Covaci restaurant) (Source: personal archive); (**C**) Crap lipovenesc/Old Believer's carp (traditional Danube's Delta fishermen's carp with fresh garlic dressing and polenta. (Lacrimi si Sfinti restaurant) (Source: personal archive).

The third cluster, in blue, presents popular variants of the main types of dishes identified by the first two clusters (e.g., ciorbă perişoare—meatball sour soup; ciorbă pui—chicken sour soup; ceafă porc—grilled pork neck; cârnaţi casă—house sausages), popular accompagnements (e.g., cartofi prăjiţi—fries; usturoi—garlic; smântână—sour cream), and other typical dishes (e.g., ficăţei—chicken liver) or traditional ways of cooking (e.g., pui ceaun—deep-fried chicken in a cast iron pot, ciorbă în pâine—sour soup in bread) (Figure 11).

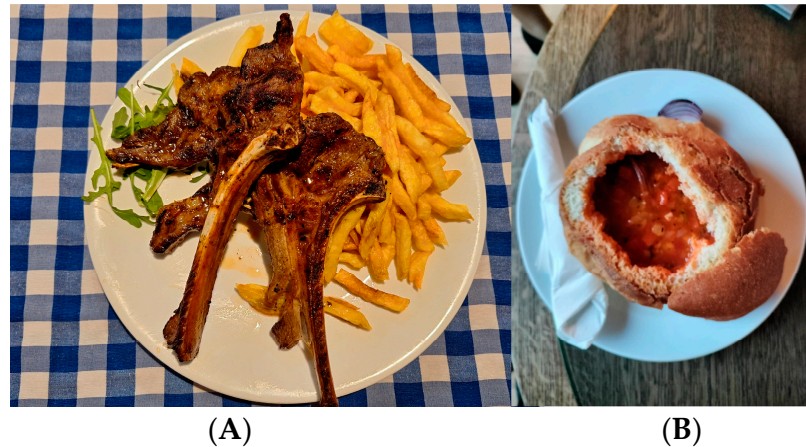

**Figure 11.** (**A**) Fries and meat dish Bodega "La Mahala". (Source: personal archive); (**B**) Sour soup in bread (Manuc's Inn restaurant) (Source: personal archive).

A fourth associated cluster, in yellow, and mainly associated with the red one, is represented by popular pork dishes (e.g., cotlet porc—pork loin; ciolan afumat—smoked pork knuckle with bone; cârnați—sausages), popular traditional side dishes (e.g., varză—cabbage, cartofi auriți—golden baked potatoes), starters (e.g., supă pui—chicken soup, which, compared to the one in the blue cluster, is not sour and is usually made with home noodles), popular desserts (e.g., înghețată—ice cream), and traditional cooking instruments used to prepare numerous dishes (e.g., cuptor—oven) (Figure 12).

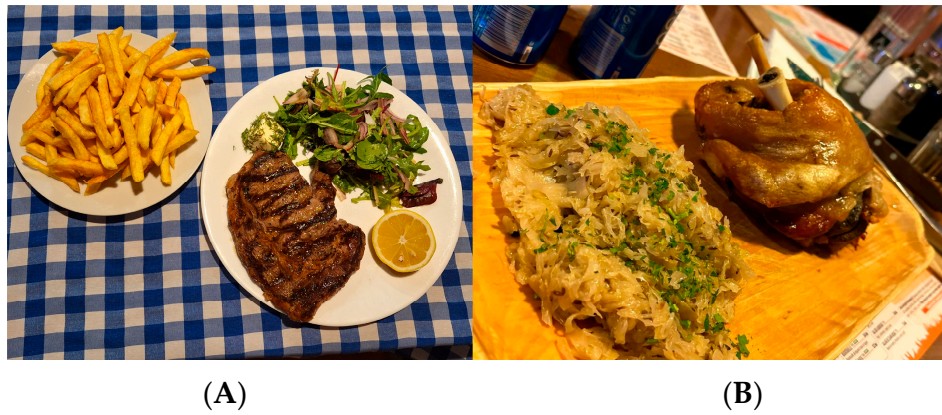

**Figure 12.** (**A**) Pork loin accompanied by fries. Bodega "La Mahala" (Source: personal archive); (**B**) Smoked pork knuckle with bone and cabbage in the Old Center of Bucharest. (Source: personal archive).

Figure 7 clearly shows elements of authenticity, reflected by the menus of Romanian-themed restaurants in the center of Bucharest: the existence of meat courses traditionally based on pork and chicken; side dishes based on potatoes, cabbage, beans, or mushrooms; and starters centered on sour soups (ciorbă) and raw products (e.g., cheese, pastrami, eggplant, zacuscă). The rural origins of dishes are reflected by the mention of ancestral simple cuisine instruments (e.g., cast iron pot, oven) or old techniques for cooking (e.g., spit roast, grilling, frying) or by direct words and references to peasant cuisine (e.g., țărănească—peasant; platoul lui Moş Ghiţă—Father/Old Ghiţă platter). Geographical references are also present in dishes' denominations and/or descriptions (e.g., traditional dish from Bucovina—Caru' cu Bere Restaurant; Maglavit tomato soup—Lacrimi şi Sfinţi; oltenian salad/salad from Oltenia—Curtea Berarilor), underscoring the importance of local regional ingredients as a guarantee for food taste and quality [105].

Customers respond to certain mental stimuli, such as original and familiar words or particular descriptions and elements that evoke emotions and represent reference attributes of persuasive strategies for food choices [106]. Therefore, an element of authenticity present in the menus and particularly perceived by autochthonous customers is represented by the humorous terms and regional or archaic descriptions of dishes (e.g., pârjoale cuvioase—translated as pious croquettes, means a vegan Romanian chiftea which normally is made of meat, except for Lent period (equivalent of köfte); ciulama republicană—translated as republican ciulama; cârnați acordați—well-tuned sausages; plăcintă sinceră—"honest" pie in the menu of Lacrimi și Sfinți). Other examples in the menu of Hanul lui Manuc are platou giugiuc/giugiuc platter (giugiuc being an old archaic Turkish term known in southern Romania as beautiful) and borș de curcan fanariot—Phanariot turkey sour soup (re-calling the Phanariot regime established for the principalities of Moldavia and Wallachia at the end of the 18th century and beginning of the 19th century under Ottoman administration). Customers can also try mușchiuleț de porc lucrat/worked pork tenderloin, in the humorous sense of body-built pig muscle, from the City Grill Covaci menu.

Another original characteristic of Romanian restaurants' menus is the usage of diminutives, usually culturally used for dear persons and objects and employed in this case for popular dishes. Food diminutives constitute a cultural characteristic of Romania in general, reflecting an emotional connection with familiar, home-made courses. Within the restaurant menus of the 22 restaurants, both diminutives and non-diminutive forms appear (e.g., cârnați—cârnăciori; sarmale—sărmăluțe, mici—mititei, mămăligă—mămăliguță, etc.), with an obvious prevalence of the first category (sărmăluțe is used in 88.9% of cases; ciorbă de văcuță instead of vacă is used 100% of the time; cârnăciori instead of cârnați is used in 82.1% of cases; and mămăliguță is used instead of mămăligă in 66.7% of cases) (Figure 13). The term mititei (used in 36.3% of cases) is a particular example representing a diminutive of another diminutive (mici means small in Romanian and is, therefore, a substantivized adjective).

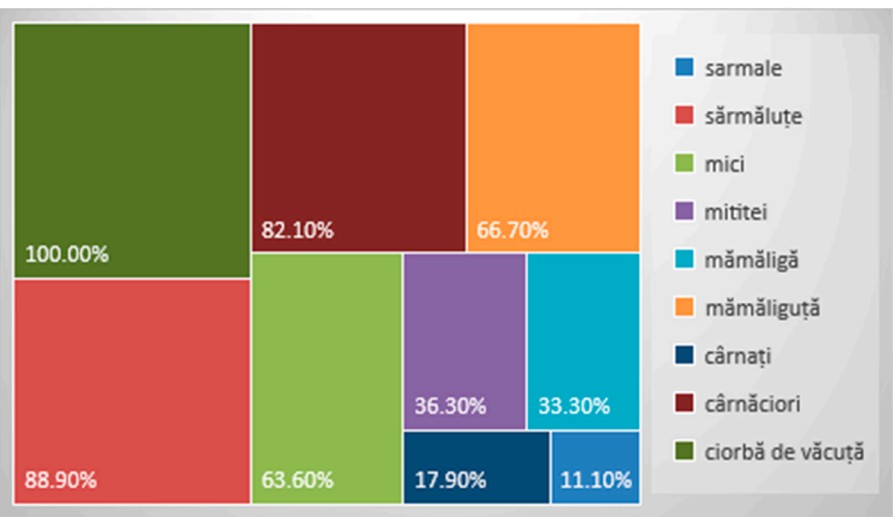

**Figure 13.** Common diminutives in Romanian cuisine, present in restaurants' menus.

### 4.3. Perception of Restaurant Authenticity

The analysis of endogenous perception focused on determining the presence of ethnic themes in the studied restaurants, which differentiate them from competitors and lead to the creation of cultural experiences [76,106].

As underlined by the analysis of the word cloud in the first part of our research, restaurants' names are an incentive for consumers, and the names of Romanian-themed restaurants in the center of Bucharest often display ethnic or historic connotations. They

trigger memories of famous personalities (e.g., Manuc, Covaci, Nenea Iancu, Burebista), make reference to inns or breweries located on ancient commercial routes (e.g., Hanul lui Manuc, Curtea Berarilor/Brewers' Court, Hanul Berarilor/Brewers' Inn, Caru' cu Bere/The Beer Wagon), induce the idea of nostalgic home-made food (e.g., La Mama), or contain more subtle references to Romanian historical background and religion (e.g., Mahala, Bodega "La Mahala", Lacrimi și Sfinți).

*"People often associate the name of a restaurant with one of its characteristics. Many times, it happened to attract tourists who went out on a simple walk through the Old City, and when noticing our logo, they stopped and exclaimed: Oh, Caru' cu Bere, I read about it. The name and the fact that the inn is well preserved all play an important part in reminding us of history, which is why tourists are interested in visiting it and trying the food."*

Employee, Caru' cu Bere

*"The medieval Romanian-specific name is very attractive for tourists; they are attracted in the first place to find out the story behind the name."*

Employee, Excalibur

According to interviewed employees, traditional Romanian restaurants comprise rural authentic elements (e.g., furniture, ornaments, pottery used for cooking and eating) that often help them recreate a vintage atmosphere, recalling memories of old inns and breweries or attracting clients through symbolic elements from legends associated with the unit's name (Figure 14).

Tourists are attracted

*"because of the traditional Romanian menu served in the location and the wooden traditional ornamenting elements, but also the pleasant atmosphere."*

Employee, Curtea Berarilor

*"We have a Romanian menu and create a traditional atmosphere; our restaurant has ornamental elements like an old peasant house from the past."*

Employee, Burebista

Explicit references to menus and descriptions of dishes confirm results and observations made in the second phase of our research and reiterate, as in the case of other studies focusing on ethnic cuisines, the fact that menus, through the original description of dishes, ingredients, and geographic origins, and through the preservation of old recipes, are perhaps "the most straightforward marker of authenticity" [107,108]. On a market numerically dominated by the presence of international restaurants that also offer popular Romanian dishes, traditional elements are underlined by the variety of dishes and techniques of cuisine.

*"The Romanian menu and the quality of the offered services… There are great food plateaus."*

Employee, Curtea Berarilor

*"Our recipes are original…. We have many foreign tourists due to the Romanian-specific traditional dishes that are selling well."*

Employee, City Grill

*"The atmosphere, the traditional food—our food is excellent. We have specialized cookers"*

Employee, Vatra

*"The dishes offered to the customers are Romanian because we want them to remember the food from another time, the food of their grandmother from the countryside."*

Employee, Mahala

A third element reflecting the authenticity of traditional Romanian restaurants in the opinion of their employees refers to originality (Figure 14), often represented by a staged authenticity and, therefore, consisting of a mix of modern and ethnic variables [72,76].

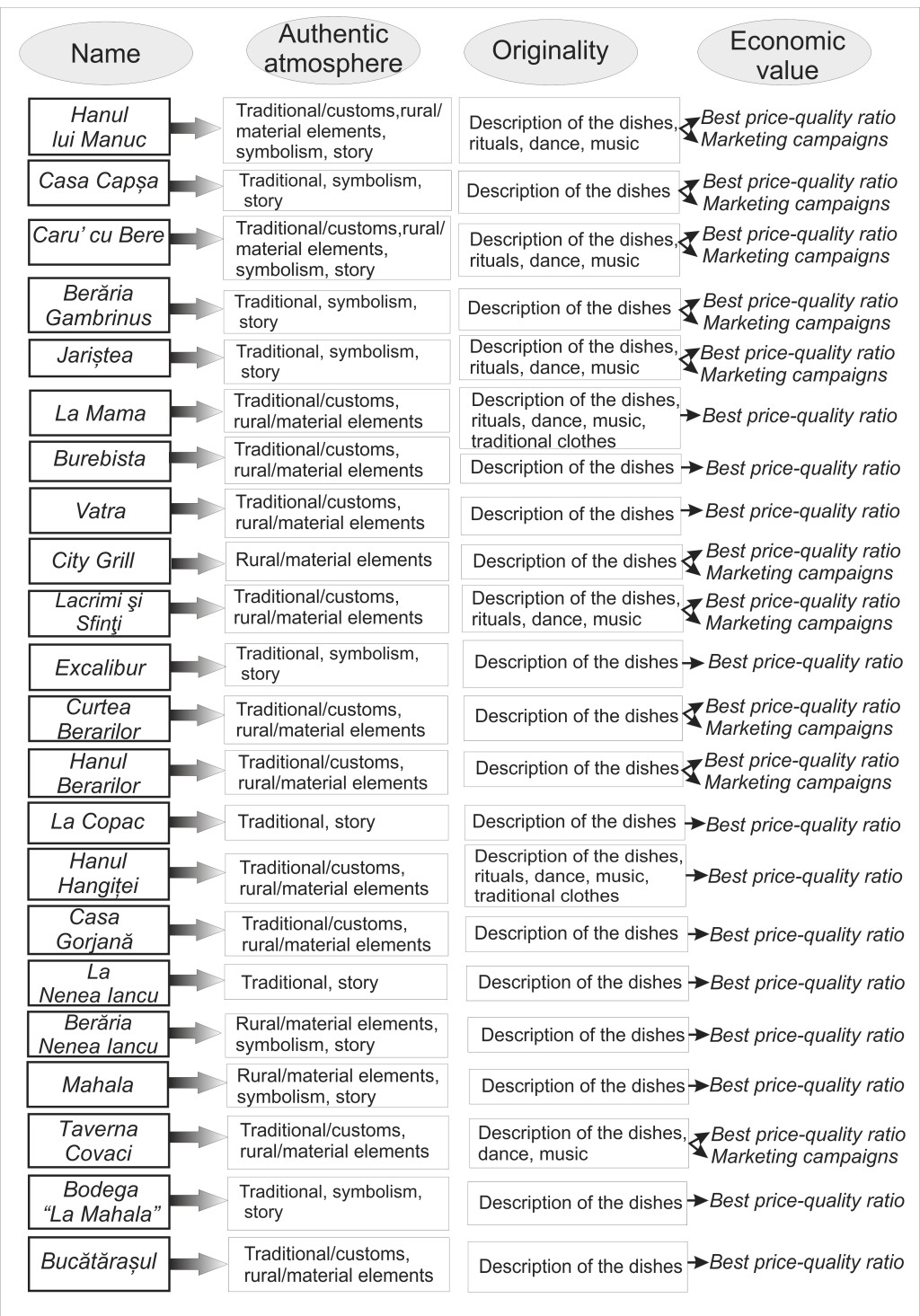

**Figure 14.** Defining authentic characteristics of Romanian-themed restaurants in the center of Bucharest from the perspective of experienced employees (based upon interview extracts).

Illusory images are created through the description of dishes, the general atmosphere and ornaments, waiters' folk outfits, music, or traditional rituals. Certain units promote themselves through authentic live music (e.g., Hanul Hangiţei) and the organization of corporate events on the occasion of great Romanian feasts (e.g., Jariştea).

> *"Since a century ago, it was oriented on this profile of traditional Romanian style. The quality of our services and the costumes and Romanian music were inspired by Romanian villages."*

Employee, Caru' cu Bere

*"We survived on this market segment because of the restaurant theme, the quality of its products and services, the clients, the atmosphere, and the position of the location, as well as the original shows."*

Employee, Excalibur

A fourth element describing restaurant authenticity as a stimulus factor for consumption is the optimization of the restaurant's economic value through a competitive price–quality ratio (Figure 14). Marketing campaigns and advertisements also represent an element used to attract both residents and tourists to become restaurant customers.

*"People constantly hear about us, including from television shows. Kera Calița is a personality of Bucharest, the soul of this place, so people come to meet her. We also promote ourselves through the parties or thematic events we organize."*

Employee, Jariștea

## 5. Discussion

An important cultural and rebranding element capable of creating positive images, traditional Romanian cuisine defines itself as a landmark of Bucharest, confirming that "gastronomy has become a significant source of identity formation in postmodern societies" [103] (p. 3), particularly in post-communist countries. Profound socio-economic and ownership transformations and entrepreneurial initiatives that occurred after the revolution in 1990 led to a spectacular growth in restaurant number and variety, as well as the appearance of gastronomy-related activities such as theme festivals and exhibitions in Bucharest.

The aim of this study was to outline the presence of Romanian-themed traditional restaurants promoting authentic local cuisine in the center of Bucharest.

In answering the first posed question, one could say that despite their relatively low number, ethnic-themed Romanian restaurants are a constant presence, representing a landmark of the local gastroscape. Several types or "generations" of Romanian ethnic restaurants with their particularities may be distinguished in the center of Bucharest, while local gastronomy has become a core element of incoming tourism packages, promising unique experiences of regional cultures [109]. Visitors increasingly demand authentic local products, in contrast to the present phenomenon of globalization of products [110], and food is an identity element [111] encountered by travelers in genuine, authentic eating places [112]. At the same time, domestic tourists and residents are mainly the ones to legitimize authentic cuisine and products in their autochthonous environment based on self-identification with the products [113], especially in EEC destinations with high importance of home cooking in food consumption [114].

Certain old authentic traditional Romanian restaurants in the center of Bucharest, such as Hanul lui Manuc, Casa Capșa, or Caru' cu Bere, became famous and attracted both authochtonous customers and tourists through their dishes and the emotional motivation induced by lived experiences. Other successful Romanian-themed restaurants in the center of Bucharest, offering high-quality services, were designed and built from scratch, emphasizing the high attractiveness of this type of cuisine in the current gastronomic landscape. However, these units account for only a small proportion of the industry, around 10% of the total number of restaurant units in the central area of Bucharest. In contrast, fast-food units or international restaurants, which better meet the global tastes and lower price standards of mass "consumers" in a more cosmopolitan European capital city that serves both local and tourist tastes, have registered impressive growth. Among stiff competition, authentic Romanian cuisine is still popular, registering a high, constant demand from residents and also from tourists, recovering their travel rates in the post-COVID-19 context. The category of mixt restaurants, not included in our study sample, offering a greater extent of typical Romanian dishes next to popular international ones in

an attempt to answer heterogenous customers in terms of gastronomic tastes is evidence in this sense.

In answering the second research question, the authenticity of ethnic restaurants in Bucharest is mainly reflected by two elements, namely food, for which the use of local ingredients and dishes or the traditional way of cooking is important, and the dining experience "usually assessed by the interior and exterior design, decorations, and music, as well as the employees' uniforms" [115] (p. 1037). For years, some ethnic Romanian restaurants have reinforced stereotypes through their standardized menus and interior design, answering consumers' expectations. Postmodern acceptance of cultural commodification for purchase and consumption and the focus on authentic tourist experiences rather justify inauthenticity as long as it succeeds in creating an aimed "enjoyable illusion" [22,73,75]. Studies showed that ethnic restaurants should make efforts to create a more authentic dining atmosphere and dishes, especially for customers displaying high cultural familiarity [115].

Maintaining the authentic character and quality of an elaborate cuisine is consequently a challenge for specialized Romanian restaurants, especially in the post-pandemic context, which emphasizes an increasing demand for on-site catering services and demonstrates the capacity of this sector to bounce back [116]. In the post-COVID period, gastronomic authenticity is being considered as a possible tourist attraction and recovery factor for Bucharest. During the COVID-19 crisis, menu originality was accentuated and became a competitive advantage. Despite the significant decrease in the number of restaurants because of the pandemic shock suffered by the entire hospitality sector, most Romanian-themed restaurants were exceptionally maintained and adapted to new consumer requests through delivery services. The visualization of semantic statistics on the text of the menus of the 22 Romanian traditional restaurants included in the study area generated four main clusters referring to the following: the most popular essential dishes from starters to desserts (e.g., ciorbă, mititei, sarmale, cârnaţi pleşcoi, papanaşi); the mămăligă (polenta) cluster as an omnipresent side dish for most main courses, traditionally replacing bread; popular variants of dishes identified in the first two clusters (e.g., ciorbă de perişoare—meatball soup, cârnaţi—sausages); and popular meat dishes (e.g., pork loin, smoked pork knuckle with bone) or side dishes usually accompanying meat in main courses (e.g., cabbage, baked potatoes). The authentic character of dishes is also underlined through their descriptions, which refer to cuisine instruments (e.g., cast iron pot, oven) and techniques (spit roast) or the use of diminutives, archaic regional terms, or humorous expressions.

In considering the third research question in this study, surpassing the opinion of those having hands-on experience about their unit's authenticity as a competitive advantage in the market was also important in the post-COVID context and represents a valuable input for the entrepreneurs in this domain and for the existing scientific literature. Addressing interviews of experienced employees was also found to be particularly useful from the exploratory perspective of this research and constituted an original, innovative approach to this topic, which is often studied from the consumers' perspective. Tourists' opinions shared on social media underscore rather subjective experiences and sometimes damage the destination's image [117]. In the opinion of interviewed experienced employees, restaurant names with historical, cultural, or emotional connotations, symbolic elements (e.g., furniture, ornaments, pottery) that project customers into the rural Romanian village atmosphere or into different historical epochs, the general atmosphere (often created by traditional music and staff clothing), and mainly traditional old recipes and the quality and quantity of food as well as the price determine an authentic experience that generates customers' satisfaction and loyalty.

## 6. Conclusions

In conclusion, Romanian cuisine, displayed by Romanian-themed restaurants in the center of Bucharest, represents a landmark of this capital city, contributing to defining its foodscape in the post-communist era for both local consumers and tourists. Gastronomy is

part of Bucharest cultural life and is an essential element of hospitality, shaping the image of the city also during themed events such as festivals or themed fairs.

Restaurants in Bucharest have continuously increased and diversified their offerings in the last three decades, and the need for authenticity and restaurant brands for traditional local cuisine, essential in tourism encounters for both international and domestic visitors, is growing.

Typical Romanian dishes are displayed next to popular international ones on the menus of a whole range of restaurants in the center of Bucharest in an attempt to answer the important demand for these products and face stiff competition on the market. As shown in our study, they are dominated by courses using local ingredients and traditional ways of cooking and characterizing slow and ethnic food cuisine in Southeast Europe such as stews and slow-cooked garnishes and accompaniments for meat-based main courses, chunky soups, and sour soups as starters or baked and fried desserts.

Experienced staff working in the traditional-themed Romanian restaurants in the center of Bucharest confirmed rustic rural decor and mainly traditional old recipes as essential parts of the authentic experience created by their unit.

These results represent a useful exploratory perspective on a less researched topic, essential for authoctonous hospitality in the light of increasing international tourist venues for Bucharest and might be of interest for both scholars and practitioners. The novel methodological approach allowing the visualization of autochtonous authentic foodscapes fills in a research gap in the scientific literature on this topic and might be further replicated in other contexts or for similar research issues and databases.

## 7. Implications, Limitations, and Future Research

In terms of theoretical implications, the current paper fills a research gap concerning the less studied topic of the authentic gastronomy landscape in Bucharest, mainly focusing on its traditional-themed Romanian restaurants in the center of this capital city. This study innovatively appealed to various complementary research methods to illustrate the authentic local foodscape. Different mapping techniques and semantic analysis were appropriately used for this topic, helping us to visualize data on restaurants and their menus and represent an innovative contribution that may fill the theoretical knowledge gap in this domain. Compared to previous studies, which focused on customers, our research innovatively appealed to qualitative interviews to explore the point of view of experienced employees in themed Romanian restaurants on the gastronomic authenticity of their unit.

Practical implications derive from the novelty of this study and also from the innovative research methods that help both scholars and stakeholders visualize menu content and find out the opinion of those having hands-on experience on the authentic elements and competitive advantages of traditional gastronomy displayed in themed Romanian restaurants in the center of Bucharest. Therefore, this study could help entrepreneurs adapt their offerings regarding traditional Romanian cuisine to meet consumer preferences and better position themselves in this very competitive market. It could be a point of interest for consumers as well, who may target specific units according to their tastes while also exploring the offerings of existing restaurant menus.

This study presents some inherent limitations imposed by the important dynamics of these units and the missing data and statistics that determine extensive field research. Some categories of units were excluded, such as restaurants in hotels or fast-food units belonging to chains with autochthonous-specific food (e.g., Ciorbe si plăcinte—soups and pies—a newly developed franchise specializing in popular traditional dishes; La Plăcinte). Either limiting their target clients (e.g., to hotel guests) or focusing on a self-service and canteen style with simple cuisine, these units do not represent a gastronomic objective for foodies and do not represent the subject of our study but may be taken into consideration by future research.

Future research and further studies may also consider a customer approach that may complement current results on the evaluation of restaurant authenticity for traditional

Romanian-themed restaurants and gastronomy in the central area of Bucharest. This is, however, a challenging approach that should take into account the different categories of consumers (e.g., residents, domestic or international tourists, people belonging to different generations or displaying different previous experiences related to traditional Romanian gastronomy), and did not match the needs and objectives of this first stage of research.

**Author Contributions:** Conceptualization, A.-I.L.-D. and M.P.; methodology, A.-I.L.-D. and M.P.; software, A.-I.L.-D. and M.P.; validation, A.-I.L.-D., M.P. and I.V.; formal analysis, A.-I.L.-D., M.P. and I.V.; investigation, A.-I.L.-D. and M.P.; data curation, A.-I.L.-D. and M.P.; writing—original draft preparation, A.-I.L.-D., M.P. and I.V.; writing—review and editing, A.-I.L.-D., M.P. and I.V.; supervision, A.-I.L.-D., M.P. and I.V.; funding acquisition, A.-I.L.-D. All authors have contributed equally to this research study. All authors have read and agreed to the published version of the manuscript.

**Funding:** This research received no external funding.

**Institutional Review Board Statement:** This study was conducted according to the guidelines of the Declaration of Helsinki and approved by the Institutional Review Board (or Ethics Committee) of Bucharest University (document no. 112/12.12.2023).

**Informed Consent Statement:** Informed consent was obtained from all subjects involved in this study.

**Data Availability Statement:** Data presented in this study are available on reasonable request from the corresponding author.

**Conflicts of Interest:** The authors declare no potential conflicts of interest with respect to the research, authorship, and/or publication of this article.

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
