# Peer review of "Authentic Romanian Gastronomy—A Landmark of Bucharest’s City Center"

_tourismhosp, doi:10.3390/tourhosp5020017_

Round 1

Reviewer 1 Report

Comments and Suggestions for Authors

Dear Authors,

I think that this paper is very original and well written. Therefore it can be accepted in its present form. Please, read again to final check the english language. 

Author Response

Dear Reviewer,

Thank you for your time and suggestions to improve our paper in view of its publication.

We followed your suggestion and had our document checked by a professional English native speaker.

Kind regards,

The authors

Reviewer 2 Report

Comments and Suggestions for Authors

Dear authors,

This is a very interesting piece of work. The article demonstrated a well-structured and well-written approach; however, I would like to offer some constructive feedback to enhance its overall quality.

Firstly, I would suggest to reorganize and improve the literature review part. This study aims to reveal the Romanian-specific elements of authentic local gastronomy for themed traditional restaurants in the center of Bucharest. In the literature review section, 2.1 and 2.2 both focus on the traditional food. I would suggest to adjust the focus into: Traditional gastronomy; Authentic traditional gastronomy; Themed traditional restaurant. This would also be in accordance with the Results section.

Secondly, in the Discussion and conclusions section (Line608-610), the authors mentions “The aim of this study was to outline, in the post-pandemic context, which represents a resetting moment for the whole hospitality industry, the presence of Romanian-themed traditional restaurants in the center of Bucharest, promoting authentic local cuisine and facing stiff competition on the market”. In particular, it highlights “the post-pandemic context”, however, this context is not introduced at the beginning of the paper, which might cause some confusion. Will the post-pandemic context make any difference to the research finding? I would suggest the authors to clarify on this to reduce the confusion.

Lastly, for the Discussion and conclusions section, I would suggest the authors to use sub-headings to more clearly categorize this section. In addition, I would suggest the authors to further enhance both the theoretical implications and practical implications.

Author Response

Dear authors,

This is a very interesting piece of work. The article demonstrated a well-structured and well-written approach; however, I would like to offer some constructive feedback to enhance its overall quality.

Dear Reviewer,

Thank you very much for your appreciative comments regarding our paper and also for your time and suggestions that helped us improve our paper in view of its publication.

Considering each recommendations we brought the following changes:

Firstly, I would suggest to reorganize and improve the literature review part. This study aims to reveal the Romanian-specific elements of authentic local gastronomy for themed traditional restaurants in the center of Bucharest. In the literature review section, 2.1 and 2.2 both focus on the traditional food. I would suggest to adjust the focus into: Traditional gastronomy; Authentic traditional gastronomy; Themed traditional restaurant. This would also be in accordance with the Results section.

Thank you very much for your suggestion. Literature review was improved and new titles were added on section 2.2 so as to include more titles from corresponding studies in Southeast Europe (Lines 186-204).

Secondly, in the Discussion and conclusions section (Line 608-610), the authors mentions “The aim of this study was to outline, in the post-pandemic context, which represents a resetting moment for the whole hospitality industry, the presence of Romanian-themed traditional restaurants in the center of Bucharest, promoting authentic local cuisine and facing stiff competition on the market”. In particular, it highlights “the post-pandemic context”, however, this context is not introduced at the beginning of the paper, which might cause some confusion. Will the post-pandemic context make any difference to the research finding? I would suggest the authors to clarify on this to reduce the confusion.

 As suggested the remark on post-pandemic context was removed from this part to avoid confusion.

Lastly, for the Discussion and conclusions section, I would suggest the authors to use sub-headings to more clearly categorize this section. In addition, I would suggest the authors to further enhance both the theoretical implications and practical implications.

Thank you for this recommendation. Discussion and conclusion part was split into several separate sections and appropriate subheadings were added. Both theoretical and practical implications were enhanced.

Thank you again for your time and valuable suggestions that helped us improve the quality of our paper.

Kind regards,

The authors

Reviewer 3 Report

Comments and Suggestions for Authors

The paper highlights the gastronomic elements that play an important role in shaping the identity and originality of the landscape in the center of Bucharest. This novel insight into gastronomy helps bridge the gap in research on this topic in the Southeast European region, particularly in relation to Romanian-themed restaurants, which serve as landmarks in the capital city. The references are appropriate and align with the research topic.

There are areas that require improvement in this paper:

    The literature review section requires additional references related to research conducted on the same or similar topics in Southeast Europe.

    In the third part of the research, the Methods section, it is essential to include a sub-chapter titled 'Place of Research.' This should encompass a description of the geographical area, accompanied by a map of Southeastern Europe indicating the position of Bucharest.

In the 'Results' section, here are the suggestions for improvements:

    In the lines 357–358, include two photos—before and after, for instance, of a street or a restaurant (assuming restitution laws permit).

    In Figure 3a, incorporate a table with percentages and the number of repetitions.

    Include at least two photos of traditional dishes after each relevant paragraph, cluster.

To ensure clarity in the 'Discussion' and 'Conclusions' sections, it is essential to separate them. Within the discussion, explain the results by addressing the predefined research questions.

Author Response

Dear Reviewer,

Thank you very much for your appreciative comments regarding our paper and also for your time and suggestions that helped us improve our paper in view of its publication.

There are areas that require improvement in this paper:

Considering each recommendations we brought the following changes:

    The literature review section requires additional references related to research conducted on the same or similar topics in Southeast Europe.

Thank you very much for your suggestion. Literature review was improved and new titles were added on section 2.2 so as to include more titles from corresponding studies in Southeast Europe (Lines 186-204).

    In the third part of the research, the Methods section, it is essential to include a sub-chapter titled 'Place of Research.' This should encompass a description of the geographical area, accompanied by a map of Southeastern Europe indicating the position of Bucharest.

 Thank you very much for your suggestion. The Sub-chapter title was added for the text describing the geographical area as well as the requested map (Figure 1).

In the 'Results' section, here are the suggestions for improvements:

    In the lines 357–358, include two photos—before and after, for instance, of a street or a restaurant (assuming restitution laws permit).

 Thank you very much for your suggestion. Photos were added accordingly (Figure 3, 4, 5, 6, 7).

    In Figure 3a, incorporate a table with percentages and the number of repetitions.

 Thank you for your suggestion, Table 2 was added accordingly.

    Include at least two photos of traditional dishes after each relevant paragraph, cluster.

Thank you very much for your suggestion. Photos were added accordingly (Figures 11-23).

To ensure clarity in the 'Discussion' and 'Conclusions' sections, it is essential to separate them. Within the discussion, explain the results by addressing the predefined research questions.

Thank you very much for your suggestion. Considering your recommendation the final section was split into several separate sections and appropriate subheadings were added. As suggested the discussion explains the results by addressing the predefined research questions.

Thank you for this solution that improved quality and readability of our paper.

Reviewer 4 Report

Comments and Suggestions for Authors

The paper “Authentic Romanian Gastronomy – A Landmark of Central Bucharest” focuses on authentic gastronomy in the context of Bucharest. Specifically, it presents the local gastronomy offered from the perspective of consumers and restaurateurs in traditional themed restaurants in the center of Bucharest. The authors used a complex and multifaceted methodological approach that considers methods such as mapping, semantic analysis, text visualization and interviews. The results show a complex gastronomic perspective of Bucharest restaurants, where consumers are attracted by specific elements of authenticity such as traditional dishes, regional denominations, local ingredients, old recipes and, based on the observations of the restaurant staff, also the atmosphere of the restaurant, originality and value for money influence the choice of local customers and tourists

The paper is interesting and useful for deepening your knowledge of traditional Romanian catering. It is well written and easy to read. I liked the topic and found it interesting and original. I would like to suggest dividing the final section (discussion and conclusions) into two or more sections, e.g. Discussion; Conclusion; Implications, limitations and future research. This type of solution should make the final part of the document more readable.

In any case, in my humble opinion, I believe that the article can be published after a small revision.

Author Response

Dear Reviewer,

Thank you very much for your appreciative comments regarding our paper and also for your time and suggestions that helped us improve our paper in view of its publication.

We were glad to read your comments on our paper and mostly the fact that you consider it of interesting and useful for readers of Tourism and Hospitality Journal.

The paper “Authentic Romanian Gastronomy – A Landmark of Central Bucharest” focuses on authentic gastronomy in the context of Bucharest. Specifically, it presents the local gastronomy offered from the perspective of consumers and restaurateurs in traditional themed restaurants in the center of Bucharest. The authors used a complex and multifaceted methodological approach that considers methods such as mapping, semantic analysis, text visualization and interviews. The results show a complex gastronomic perspective of Bucharest restaurants, where consumers are attracted by specific elements of authenticity such as traditional dishes, regional denominations, local ingredients, old recipes and, based on the observations of the restaurant staff, also the atmosphere of the restaurant, originality and value for money influence the choice of local customers and tourists

Thank you very much for above comments and remarks.

The paper is interesting and useful for deepening your knowledge of traditional Romanian catering. It is well written and easy to read. I liked the topic and found it interesting and original. I would like to suggest dividing the final section (discussion and conclusions) into two or more sections, e.g. Discussion; Conclusion; Implications, limitations and future research (line 646-808). This type of solution should make the final part of the document more readable.

Thank you very much for your suggestion. Considering your recommendation the final section was split into several separate sections and appropriate suggested subheadings were added. Thank you for this solution that improved quality and readability of our paper.

In any case, in my humble opinion, I believe that the article can be published after a small revision.

Thank you for your remark.

Thank you again for your time, appreciations and valuable recommendations that helped us improve the quality of our paper.

Kind regards,

The authors

Round 2

Reviewer 2 Report

Comments and Suggestions for Authors

I'm happy with the revisions that the authors made. I think it is qualified for publications. 

Author Response

Thank you very much for your contribution.

With gratitude,

The authors.

Reviewer 3 Report

Comments and Suggestions for Authors

Dear authors,

I am glad that you adopted the recommendations and now the work has strength for readers from the aspect of theory and practice.

Author Response

Thank you very much for the valuable recommendations.
With gratitude,
The authors.

Reviewer 4 Report

Comments and Suggestions for Authors

I appreciate the modifications made. I think that the new version is better than the previous one.

I would suggest to make more attention to text and related misprints/errors (e.g. different size of the text "Ciorbă dish in the old center of Bucharest" p. 13; "aIf" p.19).

Author Response

Thank you very much for the comments that led to the improvement of the article.
With gratitude,
The authors.